# Improving Learnt Local MAPF Policies with Heuristic Search

**Primary Keywords:** *(2) Learning; (7) Multi-Agent Planning;*

## Abstract

Multi-agent path finding (MAPF) is the problem of finding collision-free paths for a team of agents to reach their goal locations. State-of-the-art classical MAPF solvers typically employ heuristic search to find solutions for hundreds of agents but are typically centralized and can struggle to scale to larger numbers of agents. Machine learning (ML) approaches that learn policies for each agent are appealing as these could be decentralized systems and scale well while maintaining good solution quality. Current ML approaches to MAPF have proposed methods that have started to scratch the surface of this potential. However, state-of-the-art ML approaches produce "local" policies that only plan for a single timestep and have poor success rates and scalability. Our main idea is that we can improve a ML local policy by using heuristic search methods on the output probability distribution to resolve deadlocks and enable full horizon planning. We show several model-agnostic ways to use heuristic search with ML that significantly improves the local ML policy's success rate and scalability. To our best knowledge, we demonstrate the first time ML-based MAPF approaches have scaled to similar high congestion (e.g. 40% agent density) as state-of-the-art heuristic search methods.

## 1 Introduction

The increasing availability of robotic hardware has increased the importance of planning for robotic teams instead of individual agents. These multi-agent robotic systems will enable a multitude of capabilities like rapid search and rescue, exploration on Mars, and efficient warehouse management. Multi-agent systems are appealing as each robot can be cheap and relatively simple while the entire system is scalable and can achieve complex goals.

A fundamental problem with multiple robots is figuring out how each robot should move. Without careful consideration, robots could be stuck in a deadlock or collide with each other. Multi-agent path finding (MAPF) research focuses on developing algorithms for finding collision-free paths for a team of agents to reach their target locations in an efficient and safe manner. Although there are several possible approaches to tackle MAPF, the vast majority of these MAPF methods are heuristic search-based methods. These methods have optimality or bounded suboptimality solution guarantees and can solve long-horizon MAPF problems. However these methods typically trade off solution quality with compute time and require a centralized planner.

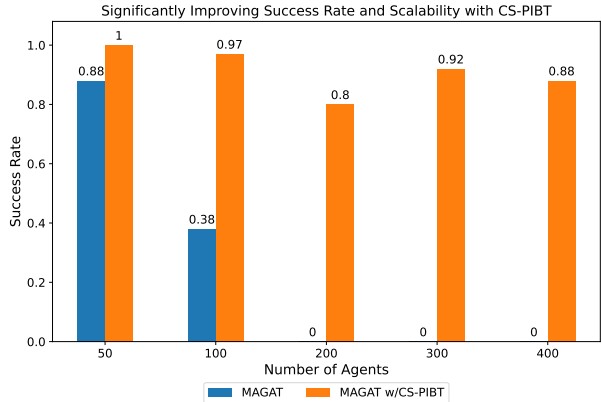

Figure 1: We compare running MAGAT (Li et al. 2021) with its default collision shielding (blue) vs running MAGAT with our PIBT collision shielding (orange). MAGAT is a learnt local policy that predicts a one-step policy per agent that could lead to collisions and therefore requires collision shielding to prevent collisions. PIBT (Okumura et al. 2022) is a heuristic search technique for solving MAPF. We see that using the exact same learnt model with our PIBT-based collision shielding dramatically improves performance and scalability without any additional training or information.

Machine learning (ML) approaches which learn policies for each agent are appealing as these could be decentralized systems which scale well while maintaining good solution quality. Current ML-based MAPF work methods are starting to scratch the beginning of this potential. State-of-the-art ML MAPF approaches learn "local" policies that take in local observations and output a 1-step action distribution. However ML in general struggles with long-horizon planning and low-error situations; both are critical in MAPF which requires long-horizon planning across many agents with very little room for errors which can cause collisions or deadlock. Thus recent ML-based MAPF works have been proposed for grid-worlds but currently do not reach the standards of state-of-the-art heuristic MAPF solvers.

Our main idea is that we can boost learnt MAPF approaches by using it with heuristic search methods. When using learnt policies that could predict actions which cause agents to collide, current ML approaches use a naive "collision shield" that replaces collisions with deadlock (see Fig-

ure 2). We instead show how we can use PIBT (Okumura et al. 2022) (a heuristic search method) as a smart collision shield during execution. We demonstrate this approach significantly boosts performance rather than solely using the learnt policy. We then show how we can more closely integrate learnt local MAPF policies with LaCAM (Okumura 2022) (another heuristic search method) to further improve performance. We explore other neural network agnostic approaches of combining a learnt policy with PIBT and LaCAM and show significant improvement in success rate and costs over the learnt policy by itself.

Our goal in this paper is *not* to show that ML approaches for MAPF are superior to classical heuristic search approaches. Our objective instead is to show that given a learnt local policy, we can use heuristic search based model-agnostic methods to significantly improve the performance of these policies. In regards to whether ML-based approaches or heuristic search approaches are better, we offer a more nuanced view. We demonstrate in our experiments that given existing strong 2D heuristics (i.e. backward Dijkstra's), current classical heuristic search approaches are extremely strong and will likely outperform ML approaches in 2D MAPF. However given imperfect heuristics, we see that ML approaches can actually outperform certain heuristic search methods. Section 4.4 discusses in further detail where learning may be applicable given our findings.

Overall, the main point we attempt to show in this paper is that ML methods for MAPF should fully leverage heuristic search. Doing so can substantially boost success rates and scalability. Succinctly stated, our main contributions are:

1. Creating a "smart" collision shield using PIBT that post-processes learnt policy conflicts instead of freezing conflicting agents.

2. Showing a neural-network agnostic framework for using a learnt 1-step policy with PIBT/LaCAM for full horizon planning to enable theoretical completeness and boost success rate, and experimenting with several variants.

## 2 Related Works

There are many different approaches for MAPF which range from optimization, heuristic search, and machine learning. We first define MAPF and then focus on the relevant heuristic search and machine learning approaches.

### 2.1 MAPF Problem Formulation

We describe the classic single-shot 2D MAPF. Here, we are given a known gridworld with free space and obstacles, and a single start-goal pair for each agent. Each agent can move in its 4 cardinal directions or wait for a total of 5 different actions per discretized timestep. Our objective is to find a collision free space-time path for all the agents to reach their goals without obstacle conflicts, vertex conflicts (two agents at the same location at the same time), or edge conflicts (two agents swapping locations across consecutive timesteps). In addition to finding a collision-free path, we hope to find efficient collision-free paths which minimize the total flowtime (sum of each agent's path until they rest at the goal).

Single-shot MAPF is harder than lifelong (where agents immediately move to a different goal if they reach their first goal) or disappear-at-goal MAPF variants as agents need to rest at the goal location. Agents waiting at the goal requires tough reasoning for learnt models as naively stopping these agents at the goal can block later agents from reaching their goal locations. Our framework is applicable to other MAPF variants too but we choose to evaluate it on the more difficult single-shot MAPF scenario.

### 2.2 Heuristic Search Approaches

Heuristic search methods aim to tackle the exponentially growing search space by intelligently leveraging the semi-independence of agents. Conflict-Based Search is a popular state-of-the-art complete and optimal framework for MAPF (Sharon et al. 2015). This technique plans for each agent individually and then resolves conflicts iteratively by applying constraints and replanning. Improvements on this foundational technique have been shown to scale up to solving hundreds of agents optimally or bounded-suboptimally (Barer et al. 2014; Boyrasky et al. 2015; Li, Ruml, and Koenig 2021; Sharon et al. 2013; Li et al. 2020a, 2019). These methods typically have good solution quality but computationally scale poorly as the number of agents increases.

**PIBT** Recently, PIBT and its extensions (Okumura et al. 2022; Okumura 2022) have shown how greedy heuristic search methods can scale extremely well at the expense of solution cost. At its core, at each timestep, PIBT has each agent greedily attempt its best action by following its individual best path. If two agents have actions that would lead to conflicts, the higher priority agent has precedence and the lower priority agent must attempt its second-best action. This procedure repeats until a conflict-free set of actions is found, if not the first agent is forced to attempt its second-best action (and logic repeats accordingly). PIBT interleaves greedy 1-step planning and execution but is still effective in long horizon MAPF tasks. PIBT uses backward Dijkstra's heuristic estimates to determine what actions are best (i.e. the action leading to the state with the least heuristic estimate is the best).

**LaCAM** LaCAM (Okumura 2022) builds on PIBT by using it as a successor generator within a Depth-First Search (DFS) of the joint-configuration space. We describe LaCAM in a simplified way to get the main idea across, but note that LaCAM is more nuanced. Given $N$ agents, imagine running a "joint-configuration" space DFS. Specifically, given the initial configuration, we generate all possible valid neighboring joint-configuration successors, pick one we haven't seen, and repeat. Note that a valid joint-configuration successor will move agents by only one step (or wait) and will not have vertex, edge, or obstacle collisions. In 2D MAPF with each agent having 5 actions, this means $N$ agents can generate up to $5^N - 1$ new successors (the minus 1 as all agents waiting will not results in a new configuration). This approach is clearly not scalable to many agents due to the exponential number of successors.

LaCAM intelligently bypasses this issue by generating an exponential number of successors lazily. The key insight is that given a joint-configuration $J_A$, we must generate all possible successors, but we can do this sequentially as we encounter them rather than all at once. They do this by

employing lazy constraints where each constraint specifies
that an agent should be at a specific location. Therefore, if
we encounter $J_A$ multiple times when back-tracking in the
DFS, we will require different agents to be at different lo-
cations and generate different joint-configuration successors
that satisfy these constraints. Figure 2 shows an example
where DFS goes to the left, exhausts successors, and then re-
visits the start configuration and lazily generates a new suc-
cessor to the right (e.g. by constraining the orange agent to
be in the middle). In the limit of time and memory, LaCAM
explores all possible successors of $J_A$. It is therefore cru-
cial that the configuration generator be fast while satisfying
the constraints. LaCAM found that PIBT performed well as
a configuration generator compared to other MAPF meth-
ods. Since LaCAM eventually explores all configurations,
LaCAM is theoretically complete. In practice, LaCAM im-
proves success rate over PIBT as the DFS over configura-
tions allows getting out of local minima (e.g. deadlock). La-
CAM is extremely effective on existing 2D benchmarks.

## 2.3 Machine Learning Approaches

Machine learning approaches typically attempt to learn local
1-step policies for each agent which they then execute in
parallel to solve the MAPF instance. Each local policy is
fed a local field-of-view of the map, nearby agents, and goal
representations, and outputs a probability distribution over
the 5 possible actions.

PRIMAL (Sartoretti et al. 2019) is a foundational machine
learning method that uses reinforcement learning and super-
vised learning to learn local policies. PRIMAL2 (Damani
et al. 2021) improves the observation inputs from PRIMAL
to handle mazes by automatically annotating potential bot-
tlenecks. SCRIMP (Wang et al. 2023) uses the same frame-
work but replaces the complicated PRIMAL2 inputs with
extremely small 3x3 observations and inter-agent communi-
cation via a modified transformer.

GNN (Li et al. 2020b) is a popular approach that solely
uses supervised learning to learn a local policy. They use
a Graph Neural Network (Gama et al. 2019) for communi-
cation and symmetry breaking across agents. MAGAT (Li
et al. 2021) improves upon the message-passing neural net-
work architecture in GNN. GNN introduces (and MAGAT
also uses) a "collision shield" which takes each agent's pre-
ferred action and executes it if collision-free or freezes the
agent if it would cause a collision. Other works, e.g. PRI-
MAL and PRIMAL2, do not explicitly define their collision
shield but use similar implicit shielding.

A key problem highlighted by most ML works is dead-
lock between agents. PRIMAL's results show how deadlock
can commonly occur when agents rest at their goal location.
PRIMAL2 specifically aims to learn conventions to decrease
deadlock in maze structures. SCRIMP uses their learnt state-
values (an additional output of their model apart from their
local policy) to calculate agent priorities that they use to
prioritize agents in deadlock, and show that this improves
performance over naive collision shielding. We qualitatively
noticed how MAGAT encounters deadlock as well. Our in-
sight is that we can resolve local deadlock using heuristic
search methods on top of the learnt policy predictions.

# 3 Improving Learnt Local Policies with Heuristic Search

Machine learning MAPF policy methods aim to learn a local
policy that each agent runs in a decentralized manner. Theo-
retically, these techniques should be able to scale well to an
increasing number of agents as the time should be roughly
constant regardless of the number of agents. However exist-
ing learnt methods iteratively plan and execute 1-step poli-
cies that get stuck in deadlock or live-lock, lack full hori-
zon planning and theoretical completeness guarantees, and
in practice struggle with poor success rates and scalability.

## 3.1 Collision Shielding: Handling 1-step Conflicts with PIBT

One fundamental issue with using learnt 1-step policies is
that it is possible for the agents' learnt policies to choose ac-
tions that lead to collisions. Figure 2 shows how two agents
could choose to move into the same empty cell and lead to
a collision if unchecked. Existing 1-step MAPF ML works
avoid this by post-processing the outputs to check if the ac-
tions are collision-free, and freeze agents that propose con-
flicting actions. GNN (Li et al. 2020b) introduces this pro-
cess as "collision shielding".

---

**Algorithm 1: Collision Shield PIBT ("CS-PIBT")**

---

**Parameters**: Current state $s^i$, action probability distribution
$p^i_{1:5}$ for each action agent $i$. Note $\|p^i\|$ is 5 as in 2D MAPF
we have 4 movements along with waiting. $p^i_{1:5}$ can be from
an arbitrary learnt model.
**Output**: Collision free actions and successor states $\forall i$

1: **procedure** GETSTRICTACTIONORDERING($p^i_{1:5}$)
2:     **return** action ordering sorted by decreasing $p^i_{1:5}$
3: **procedure** GETSAMPLEDACTIONORDERING($p^i_{1:5}$)
4:     **return** reorder action orders by $p^i_{1:5}$ biased sampling
    without replacement
5: **procedure** CS-PIBT($s^i, p^i, \forall i \in [1, N]$)
6:     $a^i_{1:5} \leftarrow$ GetSampledActionOrdering($p^i$) $\forall i$
7:     $(s^{1:N}_{new}, a^{1:N}_{best}) \leftarrow$ PIBT($s^{1:N}, a^{1:N}_{1:5}$)
8:     **return** $(s^i_{new}, a^i_{best})$ $\forall i$

---

Formally, collision shielding is a function that takes in
the current configuration and proposed action distribution
and returns the next valid configuration that avoids vertex,
edge, and obstacle collisions. Li et al. (2020b) describes the
following collision shielding process we term "CS-Naive".
Given $N$ agents in a joint-configuration $s^{1:N}$ and actions
$a^{1:N}$, we simulate $(s^i, a^i) \rightarrow s^i_{new}$ and detect collisions.
All agents with collisions are told to wait at their current
location. As the authors write themselves, this can cause
deadlock if multiple agents repeatedly propose actions that
lead to conflicts. Figure 2 shows an instance where colli-
sion shielding is necessary, and how two agents colliding
can cause other agents to be stuck in deadlock.

One critical observation is that this collision checking
does not take the full probability distribution of the agents'
proposed actions. Specifically, we either take the chosen ac-
tion or we wait, we never consider the other actions. Sup-
pose we have two agents at different locations proposing to

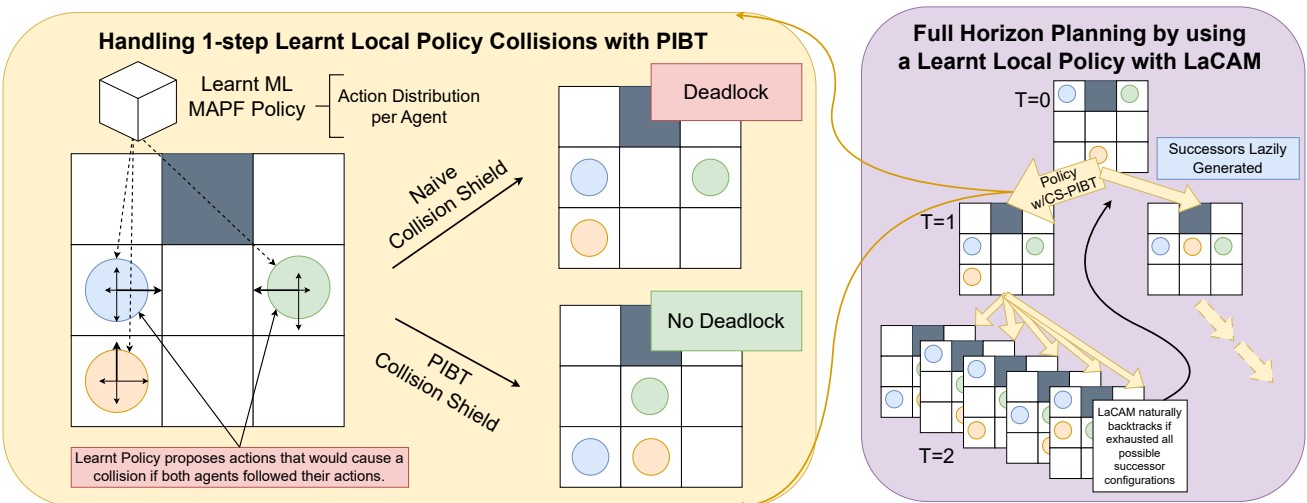

Figure 2: Given a learnt local MAPF model, we need to resolve possible 1-step collisions that might occur if we followed the proposed actions. We depict one such instance where blue and green would collide into each other. Existing work uses a "naive collision shield" which only uses the agents' picked actions and replaces collisions with wait actions, which can cause deadlock between agents. We propose using PIBT collision shield (CS-PIBT) to resolve 1-step collisions without deadlock. Note that CS-PIBT uses the entire action distribution of the agent. To enable full horizon planning, we can use the LaCAM framework with the learnt policy with CS-PIBT as the configuration generator as defined in Okumura (2022). LaCAM in essence conducts a DFS over the joint-configuration space, enabling it to escape local minima by backtracking and improving success rates.

go to the same location, what if we could let one go there and the other agent pick its second best action? This could significantly reduce idling and deadlock. But now the question arises of which agent can attempt its primary action and which agent needs to try its second action. Priority-based techniques naturally solve this question by assigning priority to agents and having the better priority agent have precedence. We thus want a single-step priority-based approach that takes in a preference of actions and returns a valid configuration. PIBT does exactly this.

Concretely, instead of solely taking in the single proposed action per agent, we propose feeding the entire probability distribution into PIBT along with agent priorities. PIBT naturally resolves obstacle, vertex, and edge conflicts using agent priorities, backtracking, and the full action set. We term this use of PIBT as a collision shield for a learnt policy as "CS-PIBT". CS-PIBT will always return a valid configuration as all agents waiting is a valid collision-free option. PIBT can be decentralized (Okumura et al. 2022) and CS-PIBT is decentralized similarly to CS-Naive in that only colliding agents need to coordinate. We maintain the same iterative 1-step planning and execution process as before.

One small detail is that PIBT does not take in an action distribution but rather a strict action ordering (e.g. which action to try 1st, 2nd, ... 5th). We can easily convert an action probability distribution to an action ordering by just preferring actions in order of highest probability. Interestingly, as discussed later in our experiments, we found that although this can significantly improve performance over CS-Naive, this strict ordering is a large bottleneck. We can get much larger performance benefits by converting our action distribution into an action ordering by biased sampling. Algorithm 1 describes CS-PIBT with the two possible variants.

## 3.2 Full Horizon Planning with LaCAM

The previous section describes a PIBT-based "collision shield" that handles 1-step conflicts. However, using a learnt 1-step policy with this collision shield does not enable any theoretical guarantees as the 1-step policy can still get stuck in deadlock or be arbitrarily bad. Ideally, we would like to use a learnt 1-step policy in a manner that maintains theoretical solution guarantees.

Section 2.2 describes in depth how normal LaCAM works. The key observation is that LaCAM requires a fast configuration generator that can satisfy lazily added constraints, and that PIBT satisfies this. We observe that a learnt local MAPF policy with CS-PIBT can be modified to be a valid configuration generator. Specifically, in order to work with LaCAM, our learnt model with collision shielding needs to handle constraints that get lazily added. Given constraints on agents, we can easily do this by invalidating proposed actions that violate constraints and having our collision shielding only consider the valid subset (the same way as is done in PIBT within regular LaCAM). Another perspective is that instead of LaCAM using PIBT informed by a normal cost-to-goal heuristic, we can use PIBT informed by the learnt local policy action distribution.

Since the learnt policy in this approach only reorders action preferences, it only alters the order in which configurations are searched and does not prune out any configurations. Thus using a local 1-step learnt policy with CS-PIBT within LaCAM enables full horizon planning with the same completeness guarantee as LaCAM. Using LaCAM does require a centralized search effort wrapping over the decentralized model with CS-PIBT, but this centralized structure is significantly weaker than other MAPF methods like EECBS (Li, Ruml, and Koenig 2021) or MAPF-LNS2 (Li

et al. 2022) which requires iterations of sequential replanning of agents. We note there currently exists no decentralized MAPF method that ensures completeness.

### 3.3 Combining a Local Policy with a Heuristic

As mentioned, one perspective of using a learnt model with CS-PIBT as a configuration generator in LaCAM is that we are using regular PIBT informed by the learnt local policy $\pi$ instead of by a cost-to-goal heuristic $h(s)$. This perspective implies that some middle ground is possible; we can use PIBT informed by both the heuristic as well as the learnt local policy. This can be done in various manners.

Concretely, normal PIBT from $s$ picks the action $\min_a h(s')$ where $s'$ is a successor state of $s$ after applying $a$. In 2D MAPF, there are many instances where two actions are tied for the minimum, and Okumura et al. (2022) shows that tie-breaking randomly is important for good performance. On the other hand, our local policy with PIBT collision shielding picks the action $\max_a p_\pi(s)$. Note that there is a unit mismatch, i.e. heuristic values vs probabilities between the two objectives.

Our first observation is that instead of tie-breaking randomly in PIBT between two equally good $h(s')$ states, we can tie-break using the learnt policy action preferences. This bypasses the unit mismatch and we thus pick the lexicographic best action $\min_a(h(s'), 1 - p_\pi(a))$. However, this tie-breaking mechanism is not satisfactory as it only uses the learnt policy sparingly in ties. We can generalize the utility of the learnt policy by picking $\min_a h(s') + R \times (1 - p_\pi(a))$ where $R$ is a hyper-parameter that fixes the unit mismatch as well as serves as a weight on how much we want to follow our policy over the heuristic.

Altogether we have four possible ways of combining the local policy with a heuristic:

$$O_h = \min_a h(s') \tag{1}$$

$$O_\pi = \max_a p_\pi(a) \tag{2}$$

$$O_{tie-breaking} = \text{lexmin}_a(h(s'), 1 - p_\pi(a)) \tag{3}$$

$$O_{combined}(R) = \min_a h(s') + R \times (1 - p_\pi(a)) \tag{4}$$

Note, $O_{combined}(R = 0)$ equals $O_h$ and $O_{combined}(R \rightarrow \infty)$ is identical to $O_\pi$. In 2D MAPF with unit actions and a backward Dijkstra heuristic, given a state $s$, neighboring heuristic values $h_{BD}(s') \in \{h_{BD}(s) - 1, h_{BD}(s), h_{BD}(s) + 1\}$. By definition, $p_\pi(s) \in [0, 1]$. Thus $O_{combined}(R)$ with $R \in (0, 1]$ is equivalent to $O_{tie-breaking}$.

## 4 Experimental Results

We have described multiple techniques to boost the performance of a learnt local MAPF policy with heuristic search. We seek to evaluate the following experimentally:

1. How does PIBT collision shielding compare against naive collision shielding?

2. How does integrating a learnt policy with LaCAM boost performance?

3. What is the best way of combining a learnt policy with heuristic information?

### 4.1 Learnt Policies used for Evaluation

All of our techniques are agnostic to the methodology or architecture of the learnt policy. Implementing and incorporating PIBT collision shielding is straightforward given an arbitrary learnt policy as it is just a post-processing technique, so we were able to evaluate it on the state-of-the-art pre-trained MAGAT neural network. However, it is more complex to implement the LaCAM framework and we were unable to incorporate MAGAT's policy with its existing implementation. We instead trained a significantly simpler network ourselves that we were able to use within the more complex LaCAM framework. We evaluate our results on the standard random-32-32-10 map (Stern et al. 2019) as MAGAT is trained on environments with 10% randomly sampled obstacles. All results are aggregated across 25 provided scenarios and 5 seeds. For CS-PIBT and LaCAM, we used dynamic priorities as defined in Okumura et al. (2022) which assigns initial high priorities to farther agents and then sets priorities to zero when agents reach their goal.

**MAGAT** MAGAT is a state-of-the-art model that uses a graph neural network structure and supervised learning for one-shot MAPF (Li et al. 2021). MAGAT was shown to scale well given similar agent densities, i.e. they show that MAGAT trained on 20x20 maps with 10 agents (density of 0.025) is able to scale to 200x200 maps with 1000 agents (same 0.025 density) with an 80% success rate. Note that most classical MAPF methods evaluate performance by increasing the number of agents in the same map, i.e. increasing agent density. Given a 50x50 map, MAGAT evaluates performance up to 100 agents (density 0.04) and starts to show performance degradation after 60 agents (density 0.024). They do not show results past 0.04 density, implying MAGAT fails past this level. For context, existing heuristic search methods like EECBS, PIBT, and LaCAM can work in 40+% agent densities.

**Simple Learnt Policy** The objective of this simple learnt policy is to see how our techniques can improve relatively weak models. We intentionally train a simple policy to contrast MAGAT and avoid extra hyper-parameters and hyper-parameter tuning.

We trained our simple policy $\pi_s$ similar using supervised learning. Concretely, we ran EECBS on the random-32-32-10 map for 20-200 agents and collected each state-action pair as a training example. For each agent $a_i$ with (state, action), we fed in 6 local 9x9 fields of view centered at the state. The first 9x9 was the local map (0's free space, 1 obstacle) and the 2nd 9x9 was a local backward Dijkstra's heatmap (similar to PRIMAL2) guiding $a_i$ to its goal. The other four 9x9 channels were the backward Dijkstra's heatmap of the four closest agents within the field of view, with all-zero heatmaps if not enough agents. This is fairly in line with existing work.

The main changes are that our neural network is significantly smaller than all existing work (just 1 CNN layer, 2 MLP layers) and critically has no inter-agent graph neural network or communication structure. After the CNN encoding, we also add in the relative coordinates of the four closest agents within the field of view (or pad with zeros accordingly). Due to its small nature, the model we trained was

Success Rate

| Agents | $h_{Manhattan}$ | | $h_{BD}$ | | MAGAT | | | Simple Policy $\pi_s$ | | | |
|---|---|---|---|---|---|---|---|---|---|---|---|
| | PIBT | LaCAM | PIBT | LaCAM | CS-Naive | CS-PIBT | $O_{tie}$ | CS-Naive | CS-PIBT | LaCAM | $O_{tie}$ + LaCAM |
| 50 | 0 | 1 | 0.986 | 1 | 0.88 | 1 | 1 | 0 | 0.928 | 1 | 1 |
| 100 | - | 1 | 0.982 | 1 | 0.384 | 0.976 | 1 | - | 0.88 | 1 | 1 |
| 200 | - | 1 | 0.83 | 1 | - | 0.8 | 0.944 | - | 0.592 | 0.992 | 0.992 |
| 300 | - | 1 | 0.554 | 1 | - | 0.92 | 0.696 | - | - | 0.896 | 0.928 |
| 400 | - | 0.952 | 0.4 | 1 | - | 0.888 | 0.728 | - | - | 0.576 | 0.688 |
| 450 | - | 0.864 | 0.366 | 1 | - | 0.776 | 0.64 | - | - | 0.528 | 0.512 |

Average Path Cost per Agent over Successful Instances

| Agents | PIBT | LaCAM | PIBT | LaCAM | CS-Naive | CS-PIBT | $O_{tie}$ | CS-Naive | CS-PIBT | LaCAM | $O_{tie}$ + LaCAM |
|---|---|---|---|---|---|---|---|---|---|---|---|
| 50 | - | 130 | 25.7 | 25.6 | 26.7 | 24.6 | 23.7 | - | 26.0 | 26.2 | 25.2 |
| 200 | - | 396 | 34.4 | 34.7 | - | 39.7 | 31.0 | - | 34.7 | 34.7 | 33.7 |
| 300 | - | 614 | 40.9 | 41.1 | - | 59.5 | 36.4 | - | - | 41.0 | 39.7 |
| 400 | - | 904 | - | 49.3 | - | 114 | 45.2 | - | - | 49.0 | 47.3 |

Table 1: The top table compares the success rates of using different learnt local policies with our proposed methods. We include PIBT and LaCAM runs using a Manhattan ($h_{Manhattan}$) and backward Dijkstra's ($h_{BD}$) heuristic as baselines. Existing works using naive collision shielding (CS-Naive) to handle 1-step conflicts, we propose using a PIBT-based collision shielding (CS-PIBT). We see that CS-PIBT dramatically improves MAGAT's success rate and scalability from under 100 agents to succeeding over 70% with the maximum number of agents in the benchmark. CS-PIBT similarly significantly boosts our simple policy $\pi_s$ to 200 agents while $\pi_s$ with CS-Naive only succeeded up to 10 agents. The last two columns also show the significant performance benefit of incorporating a learnt policy with LaCAM. Note that $\pi_s$ + LaCAM doubles scalability over $\pi_s$ with CS-PIBT and is still only driven by the simple policy and has no access to additional information. The bottom table shows the average path cost per agent over successful instances that all methods were able to solve. We finally evaluate how combining the learnt policy with $h_{BD}$ affects performance. Note MAGAT $O_{tie}$ is MAGAT with CS-PIBT with actions ordered by $O_{tie-breaking}$, while the corresponding column for $\pi_s$ uses LaCAM with $O_{tie-breaking}$. We see that combining both together leads to better costs than using just the policies themselves.

unable to generalize to new maps so we just trained on the same map we tested on with a subset of agents and start-goals. As seen later on, the simple policy cannot succeed past 10 agents with CS-Naive. However, with CS-PIBT and our LaCAM framework, our policy can scale up to 200 and 400 agents respectively. This implicitly reveals how integrating heuristic search can potentially allow ML practitioners to learn simpler models (with potential computational benefits) rather than existing larger models.

### 4.2 Improving Scalability and Success Rate

**CS-PIBT** Table 1 evaluates the effectiveness of the collision shield, comparing CS-Naive to CS-PIBT on MAGAT and our simple policy. All the methods were given a 60-second timeout except for MAGAT which used the maximum makespan limit described in their paper. MAGAT with CS-Naive has reasonable success at 50 agents ($\approx$ agent density 0.05) consistent with MAGAT's results, but is unable to scale to more agents. Using the same model predictions with CS-PIBT, we see that it is able to have good success rate on 450 agents (the max number of agents in that benchmark). This is an almost 10x improvement in agent density scalability with the exact same model. Likewise, our simple policy is unable to solve 50 agents, and upon inspection can only solve up to 10 agents well. With CS-PIBT, it can scale up to 200 agents (over a 10x improvement in scalability). These results convincingly demonstrate that CS-PIBT can significantly improve success rates and scalability in congestion. When visualizing the different collision shields, we noticed that CS-PIBT has two main strengths over CS-Naive. First, CS-PIBT allows robust movement when several agents are grouped up in the same area, while in CS-Naive mainly agents on the borders moved while the internal ones took longer to get out. Second, CS-PIBT allows agents to more easily go through agents resting on their goal, which was common in most failures in CS-Naive for a low number of agents and CS-PIBT at higher numbers. Both these behaviors are direct results of CS-PIBT's usage of priority inheritance and taking the full action probability distribution into account. A small but crucial detail for using CS-PIBT as discussed in the next section is that the action probability should employ randomness.

One note on runtime is that the overhead of CS-PIBT is heavily dependent on the programming language and implementation of other processes. For $\pi_s$, CS-PIBT was implemented in C++ and took $0.05$ *milliseconds* per CS-PIBT call for 200 agents and had a negligible impact on overall runtime which was dominated by neural network input and inference time. However, CS-PIBT in Python for MAGAT took about $0.05$ seconds per call for 200 agents which translates to 40% of total runtime.

**The Importance of Randomness** The authors of PIBT mention how random tie-breaking plays an important part in improving performance (which we also found when testing out baseline PIBT). Interestingly, we find that employing randomness for converting the policy's action distribution into an action ordering is crucial to our CS-PIBT. Figure 3 shows the effect of using CS-PIBT using strict ordering vs sampled action ordering (see Algorithm 1). We see that strict ordering is strictly worse than sampling and that its performance boost degrades fast as the number of agents increases.

One hypothesis to explain this difference is that the model could be biased and certain actions' probabilities could consistently be higher than another, e.g. prob(up)=0.55 $>$

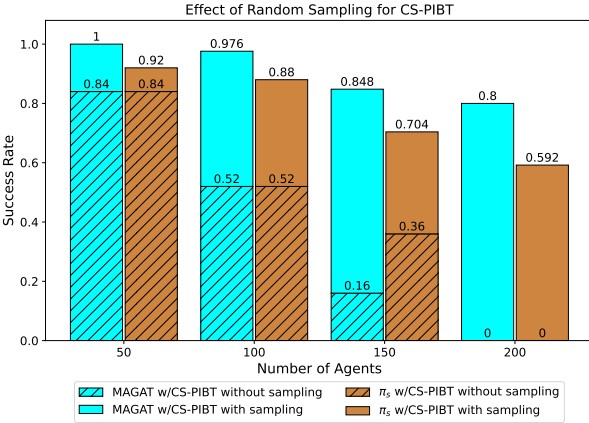

Figure 3: We plot the effect of using CS-PIBT with and without biased sampling. We see that including sampling significantly improves performance instead of always choosing actions with the highest probability first. Note that CS-PIBT with sampling allows MAGAT to scale to over 400 agents (not shown here, see Table 1) whereas it fails without sampling around 150 agents.

prob(down)=0.45. In this case, strict ordering will always try up before down, but with biased sampling we will try up before down only 55% of the time, matching the intended distribution. However, we quantitatively analyzed the action ordering distributions induced by strict action orderings and sampled action orderings and found they had very similar distributions. Qualitatively, we observed that failure instances for CS-PIBT without sampling had many livelocks where two agents alternated back and forth between the same few states, and got stuck around obstacles. We did not notice this as often with CS-PIBT with sampling, implying that sampling helps the agent overcome local minima by trying out different actions instead of repeating the same ones.

**Full Horizon Planning with LaCAM** Table 1 shows the result of running the same simple policy $\pi_s$ within the La-CAM framework. We see that LaCAM improves both the success rate and scalability over $\pi$ with CS-PIBT. LaCAM's overall framework allows searching over multiple options which enables the search to overcome local minima (e.g. deadlock) and boost success rate. These results are consistent with LaCAM's original results improving PIBT's 1-step planning. We highlight how $\pi_s$ + LaCAM significantly improves scalability as it solves 200 agents nearly perfectly while $\pi_s$ with CS-PIBT has a 59% success rate, and $\pi_s$ by itself cannot solve 15 agents.

### 4.3 Combining a Policy with a Heuristic

Figure 4 explores the different methods of combining a policy with a heuristic. We compare LaCAM solution cost against MAGAT w/CS-PIBT with action preferences sorted as described in Section 3.3. Results are percent cost increases with respect to LaCAM on successful runs, thus negative values denote cost improvements. We included an additional LaCAM baseline denoted LaCAM2 which, when given two equally good $h_{bd}(s')$, will tie-break to avoid agent

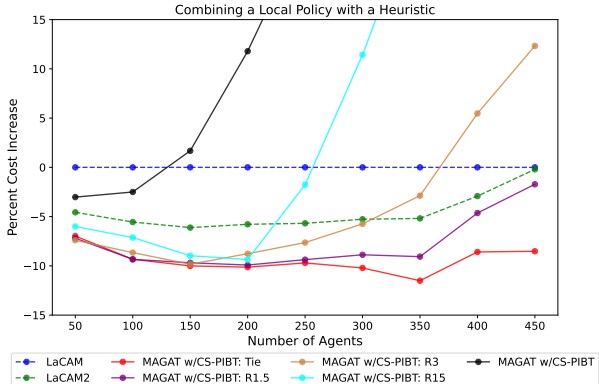

Figure 4: We evaluate different methods of combining MA-GAT's local policy with the standard backward Dijkstra's heuristic used in LaCAM. We evaluate the performance increase with respect to regular LaCAM informed by $h_{BD}$ and random tie-breaking (blue). LaCAM2 (green) tie-breaks preferring locations without other agents which was shown to improve solution cost (but reduce success rate) in Okumura et al. (2022). "Tie" (red) tie-breaks $h_{BD}$ by using MAGAT's preferences. MAGAT (black) disregards $h_{BD}$. Intermediate "R" methods (purple, orange, cyan) sort using a weighted combination of $h_{BD}$ and MAGAT's probabilities. We see that tie-breaking improves solution cost over LaCAM2 and MAGAT, revealing how combining both $h_{BD}$ and a learnt policy can lead to improvements over each individually.

collisions, which was shown in Okumura et al. (2022) to improve cost at the expense of success rate. Note that LaCAM performs full horizon planning while all the MAGAT results are iteratively 1-step planning and executing using CS-PIBT.

We observe that using just MAGAT's policy (black) leads to worse solutions than LaCAM. However tie-breaking (red), which primarily uses $h_{BD}$ and tie-breaks using MA-GAT's predictions, leads to a consistent cost improvement of around 10%. Using $O_{combined}(R)$ with different $R$ interpolates between the two behaviors, with low values of $R$ performing better. One possible hypothesis for $O_{tie-breaking}$'s better performance is that MAGAT's model simply prefers to avoid 1-step collisions. However, LaCAM2's tie-breaking mechanism does exactly this but $O_{tie-breaking}$ is still better than LaCAM2 by about 5%, with the performance improvement increasing as the number of agents increases. This implies that MAGAT is learning something more nuanced than a simple rule to get performance benefits.

We observed similar trends with our simple policy $\pi_s$ with respect to using $O_{tie-breaking}$ and $O_{combining}(R)$, except since $\pi_s$ is much simpler all costs are shifted up. $\pi_s$ with $O_{tie-breaking}$ still improves cost compared to La-CAM from 1% to around 4% with larger benefits as the number of agents increases, but it is worse than LaCAM2. $O_{combining}(R)$ with $\pi_s$ showed similar interpolation behavior as with MAGAT and similarly leads to worse performance than $O_{tie-breaking}$.

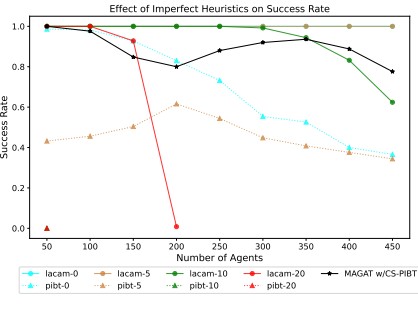
(a) Success Rate

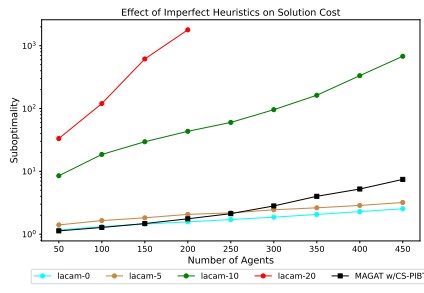
(b) Solution Cost, 0-20% $\bar{h}_{BD}$ imperfection

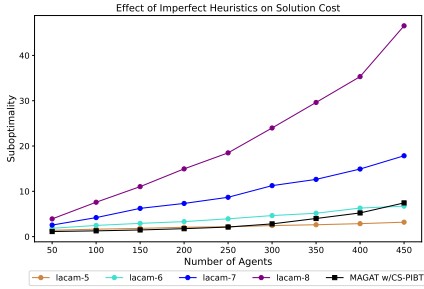
(c) Solution Cost, 5-8% $\bar{h}_{BD}$ imperfection

Figure 5: We explore the effect of imperfect heuristics on PIBT and LaCAM. Given a setting of "$K\%$" imperfection and the perfect backward Dijkstra's heuristic $h_{BD}(s)$, we uniformly sample from $e \sim [1 - K/100, 1]$ and obtain our imperfect $\bar{h}_{BD}(s) = e \times h_{BD}(s)$. (a) shows the performance with PIBT and LaCAM in $K = 0$ (cyan), 5 (brown), 10 (green), and 20 (red) $\bar{h}_{BD}$-imperfect settings. We additionally plot MAGAT which does not use $h_{BD}$ and is thus independent of the heuristic imperfections. From the success rate, we see that PIBT completely fails starting at $K = 10$ (hidden by red triangle) and LaCAM fails at $K = 20$. However, the solution cost in (b) reveals a worse picture; even though LaCAM succeeds with $K = 10$ or 20, the solutions are extremely suboptimal with LaCAM finding 100 times worse solutions. (c) highlights that LaCAM can be extremely brittle with it performing okay at $6\%$ but producing substantially worse solutions at $7\%$.

## 4.4 Should We Even Use Learnt Policies?

One observation with Table 1 is that LaCAM serves as an extremely strong baseline. LaCAM with the backward Dijkstra's heuristic ($h_{BD}$) has perfect success rate, good cost, and is extremely fast. LaCAM requires no complex training, hyper-parameters, and works on arbitrary 2D maps. All learnt MAPF works require non-trivial data collection and complex models but perform worse than LaCAM.

Our solution quality results in Section 4.3 show one possible benefit of using learnt models; we can use them with heuristic search to improve path costs. In scenarios where cost is important, these 5-10% percentage improvements could be worth the complications of learning a policy.

But what if small solution cost differences are not worth the infrastructure required for learning a MAPF policy, and we primarily care about success rate? LaCAM seems clearly superior to existing learnt MAPF models in this case.

However, one crucial assumption in 2D MAPF heuristic search methods is that we have access to an extremely strong single-agent heuristic, the backward Dijkstra's heuristic. Given no inter-agent interactions, this heuristic is perfect. Figure 5 shows that PIBT and LaCAM are extremely reliant on this heuristic. We compute a "$K\%$" imperfect heuristic $\bar{h}_{BD}(s) = e \times h_{BD}(s)$ where $e \sim [1 - K/100, 1]$. Note this heuristic is still admissible. Figure 5a shows that with 10% imperfections, PIBT fails (bottom left covered by red triangle) and 5b shows LaCAM outputs extremely suboptimal to the extent of non-usable solutions. A finer manipulation into $K$ in Figure 5c shows that LaCAM is extremely brittle to heuristic imperfections as it succeeds reasonably with $K = 6$ but produces highly suboptimal paths with $K = 7, 8$. Table 1's PIBT and LaCAM results with $h_{Manhattan}$ show similar results when using a Manhattan distance heuristic. MAGAT is not reliant on $h_{BD}$ and outperforms PIBT and LaCAM for $K \geq 7$ or $h_{Manhattan}$.

Thus in 2D MAPF, given a perfect backward Dijkstra's heuristic, LaCAM's success rates are impressive and it seems very hard for a learnt local policy to beat it. However, in scenarios where we cannot obtain a perfect or nearly-perfect backward Dijkstra's heuristic, learnt policies have the potential to outperform heuristic search algorithms. The particular 2D MAPF scenarios where we think learnt policies may be useful are instances where $h_{BD}$ cannot be computed, such as in partially observable MAPF (where only part of the map is observed), or in extreme lifelong MAPF where goals are frequently changing and the overhead for computing $h_{BD}$ becomes prohibitively expensive.

Learnt policies instead will likely excel in high dimensional state-space problems required for more realistic robot models. For example, 2D warehouse agents are constrained to move in grids at unit velocity in standard 2D MAPF formulation, but in reality, can move at angles and non-unit velocities. Realistic MAPF would then need to solve at least a 4-dimensional problem (x,y,$\theta$,velocity). Computing a perfect backward Dijkstra's heuristic or other heuristics within 6% imperfections will likely be impossible here. Learnt MAPF policies, combined with CS-PIBT or LaCAM, would be promising in these situations.

## 5 Conclusion

We showed several model-agnostic methods of improving learnt local MAPF policies with heuristic search. We first introduced CS-PIBT which is a collision shield using PIBT that takes in a policy's 1-step probability distribution and outputs valid collision-free steps. We demonstrated how this significantly improves scalability and success rate without changing the model. We then showed how we can use a learnt model with LaCAM to enable full horizon planning with theoretical completeness and in practice further boosts success and scalability. From our literature review, we have shown results for the first time where a MAPF learnt policy scales to similar agent densities (40%) of classical heuristic search methods. We hope future methods that learn local MAPF policies utilize these methods, and that more broadly, researchers work more on improving learning with heuristic search techniques.

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
