# OpenReview forum: "Improving Learnt Local MAPF Policies with Heuristic Search"
_icaps-conference.org/ICAPS/2024/Conference — ICAPS 2024_

### Official Review · Reviewer_G2mb · 2024-01-02

**Significance And Importance:** 2
**Soundness:** 3
**Novelty:** 3
**Clarity:** 3
**Overall Evaluation:** 1
**Confidence:** 5

**Weaknesses:**

0: Minor weaknesses requiring some work to be addressed for the paper to be accepted.

**Contributions Of The Paper:**

This paper proposes a Multi-agent path finding (MAPF) method by combining machine learning models with heuristic search. Specifically, they use existing works PIBT for heuristic search and LaCAM for long term view.

**Ethical Considerations:**

(1) Not Applicable: The paper does not have any ethical considerations to address

**Nomination For Best Paper:**

No

**Questions For Authors:**

Major:
•	In Abstract:
-	It sounds contradictory to me that classical solvers can find solutions for hundreds of agents but still struggle to scale to larger numbers of agents. Could you elaborate what applications and how many agents this paper is talking about. The two examples: exploration on Mars and warehouse management in Introduction seem not need so many agents.
•	In Introduction:
-	Please clarify why deadlock is a bad thing. For instance, does it affect mission planning or energy consumption?
-	‘ML in general struggles with long-horizon planning and low-error situations’ I don't agree that ML in general struggles with long-horizon planning. Reinforcement learning models consider long-horizon planning by maximizing the Q/V values. Could you elaborate a bit?
-	Please clarify what is a collision shield.
-	Please clarify why you specifically chose the two heuristic methods: PIBT and LaCAM.
-	Please clarify if the shield is generated by heuristic method instead of ML models, why it is 'smart'?
•	In Related Work:
1)	For PIBT: What if the second-best path also didn't work? In the worst case, what if all of its solutions don't work?
2)	For LaCAM:
-	Please define ‘find successors lazily’ what do you mean by lazily?
-	‘but we can do this sequentially as we encounter them rather than all at once’ Why are you sure you will encounter them later? Is it possible that in some cases, some states will never be visited?
-	In the situation above, how can you guarantee ‘In the limit of time and memory, LaCAM explores all possible successors of JA’
-	Please clarify why did you constrain the orange agent to be in the middle in Figure 2.
3)	For Machine Learning Approaches:
-	What is implicit shielding? Does it mean penalty in reward signals? Please clarify.
•	In Section 3:
-	‘However existing learnt methods iteratively plan and execute 1-step policies that get stuck in deadlock or live-lock’ In my understanding, the agents in ML methods are trained to tackle various scenarios, including escaping from deadlock. Please elaborate why getting stuck in deadlock is a problem for ML methods.
-	 ‘PIBT does not take in an action distribution but rather a strict action ordering’ please briefly explain why action ordering is needed, and what are the benefits of doing this.
-	‘We note there currently exists no decentralized MAPF method that ensures completeness’ It is unclear to me how LaCAM ensures completeness.
-	It is unclear to me what do 'action ties' and 'tie-breaking' (line 355) mean.
-	It is unclear to me how the unit mismatch is bypassed (line 365).
-	The definition of p is inconsistent. Please make sure it's p(a) or p(s). In my understanding, the correct expression should be p(s, a=a*), meaning the probability of taking action a* under state s. Please double check and modify.
•	In Experimental Results:
1)	For Simple Learnt Policy:
-	Why did you choose EECBS to generate labels?
-	‘After the CNN encoding, we also add in the relative coordinates …’ It is unclear to me what are the inputs, and how they are fed into the network.
2)	For Table 1:
-	What is the difference between 0 and failure (-)?
-	If the CS-Naive CS-PIBT did't use O_tie, which way of combining local policy and heuristics did they use?
-	Why the success rates of 400 and 300 agents are higher than that of 200 agents? And why the success rate of 400 agents is higher than that of 300 agents?
-	From Section 3.2, I understand that LaCAM is used together with policies and CS-PIBT. But it seems here that LaCAM and CS-PIBT are used separately in Table 1. Please clarify.
-	It is unclear to me what are the costs.
-	Why the average costs for 100 and 450 agents are not shown?
-	‘Note that πs + LaCAM doubles scalability over πs with CS-PIBT and is still only driven by the simple policy’ Can't see from Table 1. How could we rule out the possibility that the improvement is caused by heuristic search or LaCAM?
3)	For Section 4.1, 4.2, and 4.3:
-	Could you also show the time consumption for each step?
-	 ‘In this case, strict ordering will always try up before down, but with biased sampling we will try up before down only 55% of the time, matching the intended distribution.’ I'm really not sure about this part. If the model is trained to behave like this, it means going up can avoid collisions. Usually we add noise in action selection only in training, not in execution. Maybe adding noise in action selection can find better actions sometimes (when the model is not well trained), but will it also increase the probability of collision?
-	It is not clear to me why CS-PIBT: Tie and CS-PIBT: R1.5 are worse than LaCAM in Figure 4.
4)	For Section 4.4: this section is interesting, and I enjoyed reading this part.
-	‘we can use them with heuristic search to improve path costs.’ I understand heuristic methods find solutions by minimizing a cost function, how can ML models further improve the path costs?
-	It is unclear what is imperfect heuristic and how it is defined (line 600).



Manor:
•	In Introduction:
-	Figure 1 is not ever referenced.
-	‘Showing a neural-network agnostic’ Why the hyphen?
•	In Section 3:
-	 ‘works avoid this’ is a typo.
-	‘for each action agent i’ is a typo.
-	‘by assigning priority to agents’ I see you explained this part in the experiments, but it is better to briefly mention how you assign the priorities here as well.
-	The value of hyper parameter R is defined too late.
-	‘when R approaches to infinity, O_combine is identical to O_\pi’ I don’t agree with this statement. It is not rigorous to define importance of terms using infinity.
•	In Experimental Results:
1)	For Section 4.1, 4.2, and 4.3:
-	‘prob(down)=0.45’ please use proper notation.
-	Please define ‘percent cost increases’.

**Reproducibility:**

2: Some details are missing, but the paper still appears to be replicable with some effort.

**Strengths Of The Paper:**

•	The paper is generally readable.
•	The general idea is interesting and feasible.
•	The experiments are solid.

**Weaknesses Of The Paper:**

•	Some details are not well explained or missing.
•	Some terms are misused or used without definition.

---

> ### Author Rebuttal · Authors · 2024-01-28
>
> Please see the start of zFY6's rebuttal.
>
> Thanks for the detailed review. Due to the character limit we cannot address all questions/typos, but we will make sure the camera-ready version addresses your writing concerns. If needed we will buy an extra page (if allowed like last year) to stay within the camera-ready page limit. Here are our answers to your main questions:
>
> Abstract: We should have clarified a time bound; Classical solvers can find solutions *given enough time*, but this can take minutes. A MAPF competition this year sponsored by Amazon Robotics (https://www.leagueofrobotrunners.org/) required planning for 100s-1000s of agents in warehouses with a 1-second timeout. Thus more scalable planners are needed for these scenarios.
>
> Intro:
> - Deadlock causes failures as agents do not reach their target goal location, and is a key problem mentioned by all ML/RL-based MAPF approaches (e.g. PRIMAL, PRIMAL2, MAGAT).
> - Our CS-PIBT collision shield is "smart" in that it uses the PIBT algorithm which uses priorities and tries multiple possible actions to resolve collisions instead of just waiting which existing methods do instead.
>
> ML approaches:
> - Existing works use "implicit shielding" as in they don't explicitly state what they do but their experiments do not report any agent-on-agent collision which implies they replace collision actions with waits (which we manually verified by inspecting their code).
> - Existing ML/RL works do get stuck in deadlock even though they explicitly try to learn conventions avoiding it. PRIMAL, PRIMAL2, SCRIMP, MAGAT all mention deadlock and provided animations of failure instances visually show deadlock occurring.
>
> Experimental section: We choose EECBS as it is a high-quality but slow MAPF solver. In addition to the heuristic heatmaps, we pass in the relative location (e.g. [-2,+3]) of nearby agents.
>
> 2)
> - 0 and - are the same
> - CS-Naive and CS-PIBT just use the policy, not the heuristic.
> - \pi_s with LaCAM uses CS-PIBT internally as describe in Section 3.2, the table label was shortened for brevity as LaCAM + CS-PIBT + \pi_s was too long.
>
> 3)
> - Lines 495-501 show the time-consumption for CS-PIBT. We can add runtime of \pi_s with LaCAM.
> - Figure 4's y-axis is cost "increase", so the negative / lower values of CS-PIBT: Tie and 1.5 are better than LaCAM, not worse.
>
> 4) LaCAM and PIBT are very greedy heuristic search methods. Thus ML-models which are trained to mimic better methods can improve path costs (e.g. Figure 4).

---

### Official Review · Reviewer_nsj2 · 2024-01-20

**Significance And Importance:** 1
**Soundness:** 3
**Novelty:** 2
**Clarity:** 3
**Confidence:** 3

**Weaknesses:**

0: Minor weaknesses requiring some work to be addressed for the paper to be accepted.

**Contributions Of The Paper:**

The paper investigates how to improve learnt local policies for solving MAPF problems using heuristic search. The work exploits known techniques in MAPF to build a smart "collision shield" (CS-PIBT), shows how to perform full horizon planning through the PIBT/LaCAM, and studies different ways of combining a learnt local policy with a heuristic. The paper includes an interesting experimental analysis evaluating the performance of the proposed techniques.

**Ethical Considerations:**

(5) Excellent: The paper comprehensively addresses all of the applicable ethical considerations

**Nomination For Best Paper:**

No

**Overall Evaluation:**

-1: (weak reject)

**Questions For Authors:**

The experiments considers 10% randomly sampled obstacles and 25 provided scenarios.
Do you expect similar results with lower/higher percentages of obstacles?
I also wonder how the experimental results generalise to other types of maps that are not purely random but have some structure.
In section 4.4 the paper mentions partially-observable maps as a context in which h_BD cannot be perfectly computed. Isn't the case that partially-observable maps could also affect the learnt local policies possibly making them less accurate?

**Reproducibility:**

2: Some details are missing, but the paper still appears to be replicable with some effort.

**Strengths Of The Paper:**

The proposed techniques to improve current learnt local policies are interesting and generally effective for the considered experimental settings. The experimental analysis is useful to understand their effectiveness with respect to the naive collision shield, although it also shows that the state-of-the-art method (LaCAM) is an extremely strong baseline.

**Weaknesses Of The Paper:**

Overall, the proposed techniques are not as much effective as the state-of-the-art LaCAM using h_BD. As shown in the experiment of Figure 5, a  possible advantage for the new techniques regards solution quality. This seems limited to 5-10% and under the assumption that only an approximation of the h_BD heuristic is available. Quality of presentation should be improved. In particular Figure 1 appears at page 1 without being referred/commented.
I am not fully convincing by the discussion in section 4.4; I would have liked to see a more detailed study on cases in which h_BD cannot be exactly computed. The paper mentions 4-dimensional problems but there is no experiment in this context supporting the claim that using of learned policies performs better than standard LaCAM. Similarly for partially-observable 2D maps.

---

> ### Author Rebuttal · Authors · 2024-01-28
>
> Please see the start of zFY6's rebuttal.
>
> There is a possible misunderstanding of Fig 4 and 5 in the weakness section. Fig 4 shows that MAGAT w/CS-PIBT and h_{BD} produces 5-10% better solution costs than LaCAM with h_{BD}, when we have access to *perfect* h_{BD} heuristics. Thus, the hybrid approach has a better solution cost (although lower success rate) than LaCAM.
>
> Fig 5 introduces imperfect heuristics and reveals how MAGAT w/CS-PIBT (and without h_{BD}) outperforms LaCAM when $\bar{h}_{BD}$ is > 6% imperfect, with substantial improvements in solution quality (~5x better at 8% imperfection, 10x at 10%, 100x at 20%, etc).
>
> Q1 & Q2: Our results and videos show we get better results with higher congestion. Thus high obstacle densities or structured maps with bottlenecks (e.g. corridors, room entrances) will lead to more congestion and we expect CS-PIBT & LaCAM to help more. We included only random obstacle results as MAGAT was trained on those maps, but will try to include additional maps as requested.
>
> Q3: Learnt *local* policies, due to their local inputs, will be unaffected by partially observable maps as long as their local inputs are not bigger than their observability (which can be enforced by construction).
>
> As mentioned (lines 80-86 & Section 4.4), we agree that if h_{BD} is available, LaCAM mostly outperforms existing ML-based methods. However Section 4.4 and Fig 5 reveal that LaCAM's performance requires a nearly perfect heuristic, h_{BD}, which may not always be available, e.g. in (x,y,theta,velocity) or partially-observable MAPF. We agree that ideally we could show results on higher dimensional MAPF; unfortunately we could not find any learnt local MAPF papers that incorporate orientation and velocity as most current MAPF approaches (e.g. all our references) focus on (x,y).
>
> The main stated weakness is that the methods (e.g. MAGAT w/CS-PIBT, \pi_s w/LaCAM) are not as good as plain LaCAM. We highlight that we are proposing model-agnostic techniques, CS-PIBT and incorporating LaCAM, to improve arbitrary learnt local policies (lines 13-19, 75-80, 636-637). We do not specifically propose MAGAT w/CS-PIBT or \pi_s with LaCAM. We use them to showcase the power of CS-PIBT and LaCAM in improving model performance, and indeed showed substantial increase in scalability (2-4x for MAGAT, 10-20x for \pi_s) over the models by themselves. Improving ML with search in this manner has never been done before and we believe this has high impact for future works.

---

### Official Review · Reviewer_zFY6 · 2024-01-22

**Significance And Importance:** 3
**Soundness:** 4
**Novelty:** 3
**Clarity:** 4
**Overall Evaluation:** 2
**Confidence:** 3

**Weaknesses:**

1: Minor weaknesses that are easily fixable.

**Contributions Of The Paper:**

This work presents a novel idea to use machine learning local policies and heuristic search methods together in multi-agent pathfinding (MAPF). It's a significant step forward in solving the problems of scalability and success rate in MAPF jobs such as intelligent manufacturing environments.

The writers suggest the Collision Shield PIBT (CS-PIBT) and combine it with the Lazy Configuration Adding Machine (LaCAM) method. These techniques make ML-based MAPF methods work better, especially when there is a lot of traffic.

Extensive experiments show that the new method is much better than existing machine learning and heuristic methods in MAPF w.r.t. success rates and scalability.

**Ethical Considerations:**

(5) Excellent: The paper comprehensively addresses all of the applicable ethical considerations

**Nomination For Best Paper:**

Yes

**Questions For Authors:**

How does the method adapt to dynamic environments with high stochasticity, commonly seen in real-world applications?

How does the performance of the proposed hybrid approach compare with approaches that use solely deep reinforcement learning in MAPF scenarios?

What are the computational requirements of the proposed method, and how does it scale with the complexity and size of the environment?

Dynamic Environment Scenario: In a search and rescue operation in a disaster-struck area, the environment can change rapidly (e.g., new obstacles due to ongoing aftershocks). How would the proposed method perform in such highly dynamic and unpredictable scenarios?

Pure Deep Learning Comparison: In a complex warehouse environment with numerous autonomous robots, a pure deep reinforcement learning approach might discover novel navigation strategies not bound by the constraints of traditional heuristic methods. How does the proposed method fare against such a scenario?

**Reproducibility:**

2: Some details are missing, but the paper still appears to be replicable with some effort.

**Strengths Of The Paper:**

Integrating ML local policies with heuristic search methods like PIBT and LaCAM is a significant innovation.

The paper thoroughly evaluates its approach across different tasks, showcasing substantial scalability and success rate ​​improvements.

It can handle high-density agent scenarios and complex planning environments.

**Weaknesses Of The Paper:**

The paper mostly talks about specific scenarios, so it's unclear how well the method works in settings that aren't predictable or based on grids. Because of this, it might not work as well in real life, where things aren't as controlled or expected as in the tests.

The method depends a lot on predictive methods already out there, such as PIBT and LaCAM. Even though it's new and different, this dependence could make finding new strategies for pathfinding harder.

Combining machine learning rules with heuristic search methods might make the computations more complex and time-consuming. This makes me wonder if the technique can be used in real life, especially when making a choice quickly is essential.

---

> ### Author Rebuttal · Authors · 2024-01-28
>
> Posting a comment to everyone here to reduce duplicates across the rebuttals.
> Thanks for the review! Upon reviewing MAGAT's codebase, we found a subtle bug where agents were marked "completed" when reaching their goal but were not marked "incomplete" when they moved off it. Note our CS-PIBT is a post-processing technique implemented independently on top of their codebase and was not involved in this bug, but fixing it affected our results for MAGAT. Our updated MAGAT success rates are:
>
> | Agents | CS-Naive | CS-PIBT | $O_{tie}$ |
> |:------:|:--------:|:-------:|:------:|
> |   50   |   0.90  |    1    |    1   |
> |   100  |   0.42  |  0.97  |  0.92  |
> |   200  |     0    |  0.59  |  0.41 |
> |   300  |     0    |    0    |  0.28  |
> |   400  |     0    |    0    |    0   |
>
> Thus for over 50% success, instead of improving scalability from 50 to 400, CS-PIBT improves MAGAT's scalability to around 200 agents. This corresponds to \~20% agent density which is still significantly larger than the largest agent density we could find in existing ML MAPF work in similar maps (8% in SCRIMP). The other trends/findings with MAGAT, e.g. Figures 3, 4, 5, are otherwise unchanged. All experiments from $\pi_s$ were written on a different codebase without issues and those results are unchanged.
>
> ———
>
> Q1 & Q3: We focus on improving ML MAPF methods for deterministic environments using heuristic search methods, both sets which  currently focus on known deterministic gridworld environments motivated by warehouses where everything is known beforehand. In dynamic scenarios, it is unclear how well either framework will work. We hypothesize that CS-PIBT/LaCAM would still help learnt local MAPF policies avoid inter-agent deadlock but would not help with dynamic obstacles.
>
> Q2 & Q5: Primal, Primal2, & Scrimp are all RL approaches which explicitly try to learn novel coordination behavior via reward tuning. For example, Primal2 and Scrimp explicitly have negative rewards for "blocking" agents to mitigate deadlock. However they still fail to scale and perform similar to MAGAT, so we expect CS-PIBT and LaCAM to similarly improve local RL MAPF policies.
>
> Q3 & last weakness: The computational overhead for CS-PIBT implemented in C++ takes 0.05 *milliseconds* for 200 agents (lines 495-499), making it very practical for real-time systems. CS-PIBT is based on one-step PIBT which can plan for 100s of agents in sub-millisecond time regardless of environment size unless in tight congestion.

---

### Meta-Review · Area_Chair_MnSX · 2024-02-03

**Recommendation:** Accept (Oral)
**Confidence:** 3

**Metareview:**

This paper has received one accept, one weak accept and one weak reject. All agree that the paper is well-written, the proposed techniques to improve current learnt local policies are interesting, and the experiments were nicely conducted. While the rebuttals answered most of the questions raised by the reviewers, one reviewer pointed out after the rebuttal that the paper assumes local policies are learnt under full observability but used with partial-observability maps. It is not clear why the size of the local inputs can be enforced to be not bigger that the observability by construction. If the local input includes information that is not observable, it seems that the learnt policy cannot be used (unless the policy was explicitly learned for partially-observable maps). The authors should clarify this point in the final version of their paper.

**Ethical Considerations:**

(1) Not Applicable: The paper does not have any ethical considerations to address